# Systematic Analysis of Multiple Imaging Modalities in Infants Diagnosed with Mucopolysaccharidosis by Newborn Screening

**DOI:** 10.3390/diagnostics15080980

**Published:** 2025-04-12

**Authors:** Chung-Lin Lee, Szu-Wen Chang, Hung-Hsiang Fang, Chih-Kuang Chuang, Huei-Ching Chiu, Ya-Hui Chang, Yuan-Rong Tu, Yun-Ting Lo, Jun-Yi Wu, Hsiang-Yu Lin, Shuan-Pei Lin

**Affiliations:** 1Department of Pediatrics, MacKay Memorial Hospital, Taipei 10449, Taiwan; clampcage@gmail.com (C.-L.L.); changszuwen@yahoo.com.tw (S.-W.C.); spty871029@hotmail.com (H.-H.F.); g880a01@mmh.org.tw (H.-C.C.); wish1001026@gmail.com (Y.-H.C.); 2Institute of Clinical Medicine, National Yang-Ming Chiao-Tung University, Taipei 112304, Taiwan; 3International Rare Disease Center, MacKay Memorial Hospital, Taipei 10449, Taiwan; andy11tw.e347@mmh.org.tw (Y.-T.L.); wl01723138@gmail.com (J.-Y.W.); 4Department of Medicine, MacKay Medical College, New Taipei City 252, Taiwan; 5Mackay Junior College of Medicine, Nursing and Management, Taipei 112021, Taiwan; 6Department of Pediatrics, Tri-Service General Hospital, National Defense Medical Center, Taipei 114202, Taiwan; 7Division of Genetics and Metabolism, Department of Medical Research, MacKay Memorial Hospital, Taipei 10449, Taiwan; mmhcck@gmail.com (C.-K.C.); likemaruko@hotmail.com (Y.-R.T.); 8College of Medicine, Fu-Jen Catholic University, Taipei 242062, Taiwan; 9Department of Medical Research, China Medical University Hospital, China Medical University, Taichung 40447, Taiwan; 10Department of Infant and Child Care, National Taipei University of Nursing and Health Sciences, Taipei 108306, Taiwan

**Keywords:** mucopolysaccharidosis, newborn screening, skeletal radiography, cardiac ultrasonography, abdominal ultrasonography, early diagnosis, biomarkers, imaging assessment, lysosomal storage disease, genotype–phenotype correlation

## Abstract

**Background/Objectives:** Mucopolysaccharidosis (MPS) is a group of progressive lysosomal storage disorders affecting multiple organ systems. Although newborn screening enables early detection, early comprehensive imaging assessment during pre-symptomatic stages remains poorly understood. This study analyzed skeletal radiographic and cardiac and abdominal ultrasonographic findings in infants diagnosed by newborn screening to establish an integrated imaging assessment model. **Methods:** This retrospective study examined 277 screen-positive cases (15 MPS I, 113 MPS II, 127 MPS IVA, and 22 MPS VI) identified through newborn screening between 2015 and 2024. All patients underwent standardized skeletal radiography and cardiac and abdominal ultrasonography. Imaging findings were analyzed in conjunction with biochemical markers and clinical parameters. **Results:** Cardiac abnormalities were most prevalent in MPS I (33.3% ASD/PFO), whereas vertebral changes were more common in MPS IVA (16.5%) and MPS II (15.9%). We observed a number of significant correlations: vertebral abnormalities correlated with keratan sulfate levels, cardiac manifestations with dermatan sulfate levels, and abdominal findings with enzyme activity levels and urinary dimethylene blue ratios. **Conclusions:** This systematic analysis of multiple imaging modalities in infants diagnosed with MPS by newborn screening demonstrates that significant abnormalities can be detected during the presymptomatic stage. Correlations between imaging findings and biochemical markers provide new insights for early diagnosis and monitoring, and support implementing comprehensive imaging protocols during the initial screen-positive cases evaluation.

## 1. Introduction

Mucopolysaccharidosis (MPS) is a group of rare lysosomal storage disorders characterized by deficient lysosomal enzymes resulting in the accumulation of glycosaminoglycans throughout the body [1]. This accumulation leads to the progressive dysfunction of multiple organs, including the bones, heart, liver, spleen, and in some types, the central nervous system [2]. Without early diagnosis and treatment, patients may develop severe complications that significantly impact their quality of life and prognosis [3].

The clinical manifestations of MPS vary by type and severity. MPS I (Hurler syndrome) and MPS II (Hunter syndrome) often present with central nervous system involvement in their severe forms, with progressive cognitive decline typically beginning in early childhood [4]. In contrast, MPS IVA (Morquio A syndrome) and MPS VI (Maroteaux-Lamy syndrome) primarily affect the skeletal and cardiopulmonary systems with typically preserved cognitive function [5]. The presence and severity of neurological manifestations significantly influence treatment decisions, with severe neurological forms potentially requiring hematopoietic stem cell transplantation rather than enzyme replacement therapy alone [6].

Recent advances in newborn screening technology have made early detection of MPS possible. Since 2015, Taiwan has incorporated MPS into its newborn screening program, utilizing tandem mass spectrometry to measure enzymatic activities in blood samples [7]. This enables the identification of affected infants before symptom onset, creating a new clinical paradigm where diagnosis precedes the appearance of characteristic features. Early diagnosis is crucial for patients with MPS, as timely intervention can prevent or slow disease progression and improve outcomes [8].

Imaging studies play a vital role in the evaluation of screening-positive infants. Skeletal radiography can assess early bone abnormalities [9], echocardiography can detect cardiac valve disease and myocardial dysfunction [10], and abdominal ultrasonography can evaluate organomegaly [11]. However, the current literature on imaging findings in newborns with positive MPS-screening is relatively limited, particularly regarding integrated analyses of these imaging modalities [12].

This study aimed to systematically analyze skeletal radiographic and cardiac and abdominal ultrasonographic findings in infants identified by newborn screening for MPS to investigate the value of these imaging studies for early diagnosis and disease evaluation. The goal was to establish a comprehensive imaging assessment model to assist clinicians in making accurate diagnostic decisions and treatment plans for screening-positive infants, ultimately facilitating timely interventions that can significantly improve long-term outcomes.

## 2. Materials and Methods

### 2.1. Study Population and Design

This retrospective study included screen-positive cases referred to MacKay Memorial Hospital, Taipei, following positive newborn screening results for MPS between January 2015 and December 2024. Inclusion criteria were newborns with reduced enzyme activity detected through tandem mass spectrometry screening who were subsequently referred to our hospital for confirmatory testing and clinical evaluation [13].

### 2.2. Imaging Studies

All patients underwent comprehensive imaging evaluations including skeletal radiography, cardiac ultrasonography, and abdominal ultrasonography. Imaging studies were conducted and interpreted according to standardized protocols by experienced radiologists and specialists. All imaging studies reported in this manuscript represent baseline evaluations performed at initial assessment following referral from newborn screening. These studies were conducted as part of our standard clinical protocol for the evaluation of individuals with biochemically confirmed MPS.

### 2.3. Skeletal Radiography

Complete skeletal surveys were conducted using standardized positioning and exposure parameters. Radiographic assessment included the following:Skull morphology and thickness;Vertebral body shape and alignment;Joint morphology focusing on hip dysplasia and genu valgum;Hand and wrist structures, including carpal and metacarpal bones;Bone density and cortical thickness;Presence of a J-shaped sella turcica.

### 2.4. Cardiac Ultrasound

Transthoracic echocardiography was performed using standardized views and measurements. The following were assessed:Cardiac valve morphology and function;Left ventricular dimensions and systolic function;Right ventricular size and function;Presence of valve regurgitation or stenosis;Wall thickness and myocardial texture;Presence of pericardial effusion.

For the purpose of this analysis, atrial septal defect (ASD) and patent foramen ovale (PFO) findings were reported together, despite their different prevalence rates in the general newborn population (approximately 0.13% for ASD versus 25–30% for PFO). This combined reporting approach was adopted for the following reasons: (1) the distinction between these two conditions can be challenging in very young infants during screening echocardiography; (2) our primary aim was to establish a baseline of any cardiac involvement for future monitoring; and (3) both findings warrant similar follow-up protocols in MPS patients. Future studies with larger cohorts and longitudinal follow-up may analyze these cardiac findings separately to better understand their distinct clinical significance in MPS.

### 2.5. Abdominal Ultrasound

Abdominal ultrasound was performed after an appropriate fasting period. The evaluation included the following:Liver size and parenchymal texture;Spleen size and echotexture;Presence of hepatosplenomegaly;Gallbladder and biliary tract assessment;Kidney size and echogenicity.

### 2.6. Biochemical Analyses

Enzyme activity measurements were performed on peripheral blood leukocytes during confirmatory testing, not on dried blood spots used in initial screening. Alpha-L-iduronidase (IDUA), iduronate-2-sulfatase (IDS), N-acetylgalactosamine-6-sulfatase (GALNS), and arylsulfatase B (ARSB) activities were assayed using fluorogenic substrates as previously described [5]. Enzyme activities were expressed as μmol/4 h/mg protein, with reference ranges established from age-matched controls. It should be noted that some patients, particularly those with MPS II, may present with enzyme activity values in the lower normal range or with values that overlap with the normal reference range. This is due to several factors including the presence of pseudodeficiency alleles, milder mutations allowing higher residual activity, and the overlapping range between the lower limit of normal and affected individuals. Therefore, diagnosis was confirmed through a comprehensive approach including genetic testing, urinary GAG analysis, and clinical evaluation in addition to enzyme activity measurements.

Urinary glycosaminoglycan (GAG) levels were quantified using the dimethylmethylene blue (DMB) spectrophotometric method. The DMB assay measures total sulfated GAGs in urine through the formation of a complex between the dye and GAG molecules, resulting in a shift in the absorption spectrum that can be quantified at 520 nm. Results were normalized to creatinine concentration and expressed as mg GAG/mmol creatinine. Age-specific reference ranges were used for interpretation.

For GAG subtype analysis, dermatan sulfate (DS), heparan sulfate (HS), and keratan sulfate (KS) were quantified using liquid chromatography-tandem mass spectrometry (LC-MS/MS). These specific GAG subtypes were analyzed as they represent the primary accumulated substrates in different MPS types: DS and HS in MPS I and II, KS in MPS IVA, and DS in MPS VI. Results were reported as μg/mL with established reference ranges for each GAG subtype.

All biochemical analyses were performed at the Division of Genetics and Metabolism, Department of Medical Research, MacKay Memorial Hospital.

### 2.7. Statistical Analysis

Statistical analyses were performed using MedCalc^®^ version 23.0.9 (MedCalc Software Ltd., Ostend, Belgium). Descriptive statistics are presented as means ± standard deviations for continuous variables and frequencies (percentages) for categorical variables. Comparisons between groups were conducted using Student’s *t*-test for continuous variables and chi-square or Fisher’s exact test for categorical variables. Statistical significance was defined as *p* < 0.05.

## 3. Results

### 3.1. Patients Demographics and Baseline Characteristics

A total of 277 infants with four different types of MPS identified through newborn screening between 2015 and 2024 (Table 1) were included. The breakdown by MPS type included MPS I (*n* = 15), MPS II (*n* = 113), MPS IVA (*n* = 127), and MPS VI (*n* = 22). The cohort was predominantly male (80.9%) with a median age at initial evaluation of 2 months (range, 1–5 months).

### 3.2. Prevalence of Imaging Abnormalities (Figure 1, Table 2)

We analyzed the prevalence of imaging abnormalities across different MPS types. Overall, the percentage of patients with any imaging abnormality was: 46.7% for MPS I (7 out of 15 patients), 33.6% for MPS II (38 out of 113 patients), 29.1% for MPS IVA (37 out of 127 patients), and 27.3% for MPS VI (6 out of 22 patients).

**Figure 1 diagnostics-15-00980-f001:**
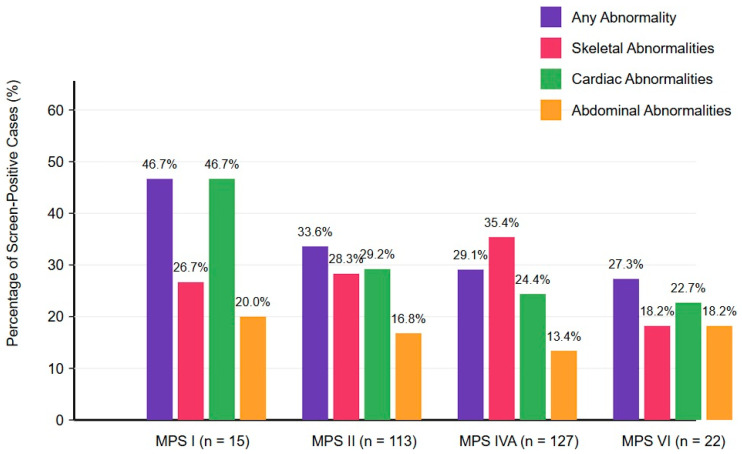
Prevalence of imaging abnormalities in different types of mucopolysaccharidosis. Bar graph showing the percentage of patients with abnormalities detected through three imaging modalities across MPS types I, II, IVA, and VI. Purple bars represent the overall percentage of cases with any abnormality. Pink bars represent skeletal abnormalities including vertebral changes, hip dysplasia, and carpal/metacarpal changes. Green bars represent cardiac abnormalities, primarily consisting of atrial septal defects/patent foramen ovale (ASD/PFO) and valvular abnormalities. Orange bars represent abdominal abnormalities, including hepatomegaly, renal pelvic dilation, and other organ involvement. Data are presented as percentages of affected patients within each.

**Table 2 diagnostics-15-00980-t002:** Radiographic abnormalities of MPS types.

Radiographic Finding	MPS I (*n* = 15)	MPS II (*n* = 113)	MPS IVA (*n* = 127)	MPS VI (*n* = 22)
**Skeletal X-ray**				
Vertebral abnormalities	1 (6.7%)	18 (15.9%)	21 (16.5%)	2 (9.1%)
Hip dysplasia	1 (6.7%)	6 (5.3%)	15 (11.8%)	1 (4.5%)
Carpal/metacarpal changes	2 (13.3%)	8 (7.1%)	9 (7.1%)	1 (4.5%)
**Cardiac Echo**				
ASD/PFO	5 (33.3%)	27 (23.9%)	23 (18.1%)	3 (13.6%)
Valvular abnormalities	2 (13.3%)	6 (5.3%)	8 (6.3%)	2 (9.1%)
**Abdominal Echo**				
Hepatomegaly	2 (13.3%)	6 (5.3%)	5 (3.9%)	0 (0%)
Splenomegaly	1 (6.7%)	4 (3.5%)	3 (2.4%)	0 (0%)
Renal abnormalities	1 (6.7%)	13 (11.5%)	12 (9.4%)	1 (4.5%)
Gastric stasis	0 (0%)	2 (1.8%)	4 (3.1%)	3 (13.6%)
Ovarian cysts (in females)	0 (0%)	0 (0%)	2 (5.5%) *	0 (0%)

* Percentage calculated based on female patients only (38 females among MPS IVA patients).

### 3.3. Radiographic Findings

Skeletal abnormalities were present in 26.7% (4/15) of MPS I patients, 28.3% (32/113) of MPS II patients, 35.4% (45/127) of MPS IVA patients, and 18.2% (4/22) of MPS VI patients. These skeletal changes represent early manifestations of dysostosis multiplex. The most common skeletal abnormality in each MPS type was as follows:MPS I: Proximal pointing of the metacarpal bones and bullet-shaped phalanges (13.3%);MPS II: Mild anterior vertebral beaking, particularly at the T12–L5 levels (15.9%);MPS IVA: Anterior vertebral beaking and posterior scalloping (16.5%);MPS VI: Vertebral body rounding (13.6%).

### 3.4. Cardiac Ultrasound Findings

Cardiac abnormalities were detected in 46.7% (7/15) of MPS I patients, 29.2% (33/113) of MPS II patients, 24.4% (31/127) of MPS IVA patients, and 22.7% (5/22) of MPS VI patients. The most common cardiac anomaly in each MPS type was as follows:MPS I: Atrial septal defect/patent foramen ovale (ASD/PFO) (33.3%);MPS II: ASD/PFO (23.9%);MPS IVA: ASD/PFO (18.1%);MPS VI: ASD/PFO (13.6%).

### 3.5. Abdominal Ultrasound Findings

Abdominal organ abnormalities were observed in 20.0% (3/15) of MPS I patients, 16.8% (19/113) of MPS II patients, 13.4% (17/127) of MPS IVA patients, and 18.2% (4/22) of MPS VI patients. The most common abdominal imaging abnormalities were as follows:MPS I: Hepatomegaly (13.3%);MPS II: Renal pelvic dilation (11.5%);MPS IVA: Renal pelvic dilation (9.4%) and ovarian cysts in females (5.5% of female patients);MPS VI: Gastric stasis (13.6%).

### 3.6. Disease Severity Classification and Imaging Correlation

To investigate whether imaging abnormalities correlate with predicted disease severity, we conducted a post-hoc analysis of our cohort. Patients were classified into three categories based on known genotype–phenotype correlations, enzyme activity levels, and biomarker profiles:Predicted severe phenotype (*n* = 78): Patients with null mutations (no detectable protein), severe missense mutations previously associated with severe phenotypes, enzyme activity < 5% of normal reference range, high urinary GAG levels (>2× upper limit of normal), or other established predictors of severe disease.Predicted attenuated phenotype (*n* = 64): Patients with specific missense mutations previously documented to associate with attenuated disease, combinations of mutations known to result in milder phenotypes (including certain splice-site variants that retain partial activity), enzyme activity between 5–15% of normal reference range, or family history of attenuated disease.Unknown/indeterminate (*n* = 135): Patients with novel mutations of uncertain significance, contradictory predictive factors, or insufficient information for classification.

### 3.7. MPS Type-Specific Severity Distribution and Imaging Findings

MPS I: Among 15 patients, 5 (33.3%) were classified as predicted severe, 4 (26.7%) as attenuated, and 6 as unknown. Imaging abnormalities were detected in 80% (4/5) of severe cases compared to 50% (2/4) of attenuated cases. Notably, cardiac abnormalities were present in both severe (100%) and attenuated (50%) phenotypes, while skeletal abnormalities were more frequent in the severe group (60% vs. 25%).MPS II: Of 113 patients, 35 (31.0%) were classified as predicted severe, 14 (12.4%) as attenuated, and 64 (56.6%) as unknown. Imaging abnormalities were observed in 48.6% (17/35) of severe cases and 35.7% (5/14) of attenuated cases. Among attenuated MPS II patients, cardiac valve abnormalities were the most common finding (35.7%), similar to reports in the literature of early cardiac manifestations in attenuated MPS II.MPS IVA: Among 127 patients, 26 (20.5%) were classified as predicted severe, 37 (29.1%) as attenuated, and 64 (50.4%) as unknown. Imaging abnormalities were detected in 53.8% (14/26) of severe cases and 24.3% (9/37) of attenuated cases. Vertebral changes were observed in both groups but were more pronounced in the severe phenotype (46.2% vs. 16.2%).MPS VI: Of 22 patients, 12 (54.5%) were classified as predicted severe, 9 (40.9%) as attenuated, and 1 (4.5%) as unknown. Imaging abnormalities were present in 41.7% (5/12) of severe cases and 11.1% (1/9) of attenuated cases. Abdominal abnormalities were predominantly observed in the severe group.

### 3.8. Biomarker Correlation with Severity and Imaging Findings (Table 3)

When stratified by predicted disease severity, the correlation between biomarkers and imaging findings remained significant but showed different patterns. In the predicted severe group, dermatan sulfate levels strongly correlated with cardiac abnormalities (*r* = 0.72, *p* < 0.001), while in the attenuated group, this correlation was moderate (*r* = 0.48, *p* = 0.008). Similarly, keratan sulfate levels showed stronger correlation with vertebral abnormalities in the severe group (*r* = 0.75, *p* < 0.001) compared to the attenuated group (*r* = 0.51, *p* = 0.004).

These findings suggest that imaging abnormalities can manifest in both severe and attenuated forms during infancy, though with different frequencies and patterns. The presence of cardiac and mild skeletal abnormalities even in patients predicted to have attenuated disease underscores the importance of comprehensive imaging evaluation regardless of predicted phenotype. 

**Table 3 diagnostics-15-00980-t003:** Imaging abnormalities by predicted disease severity.

MPS Type	Severity Category	N	Any Abnormality	Skeletal Abnormalities	Cardiac Abnormalities	Abdominal Abnormalities
**I**	Severe	5	80% (4/5)	60% (3/5)	100% (5/5)	40% (2/5)
	Attenuated	4	50% (2/4)	25% (1/4)	50% (2/4)	0% (0/4)
	Unknown	6	16.7% (1/6)	0% (0/6)	0% (0/6)	16.7% (1/6)
**II**	Severe	35	48.6% (17/35)	42.9% (15/35)	37.1% (13/35)	22.9% (8/35)
	Attenuated	14	35.7% (5/14)	14.3% (2/14)	35.7% (5/14)	7.1% (1/14)
	Unknown	64	25.0% (16/64)	23.4% (15/64)	23.4% (15/64)	15.6% (10/64)
**IVA**	Severe	26	53.8% (14/26)	46.2% (12/26)	30.8% (8/26)	19.2% (5/26)
	Attenuated	37	24.3% (9/37)	16.2% (6/37)	13.5% (5/37)	8.1% (3/37)
	Unknown	64	21.9% (14/64)	18.8% (12/64)	17.2% (11/64)	10.9% (7/64)
**VI**	Severe	12	41.7% (5/12)	25.0% (3/12)	33.3% (4/12)	25.0% (3/12)
	Attenuated	9	11.1% (1/9)	11.1% (1/9)	11.1% (1/9)	0% (0/9)
	Unknown	1	0% (0/1)	0% (0/1)	0% (0/1)	0% (0/1)

Note: Patients were classified as having predicted severe phenotype based on null mutations (no detectable protein), severe missense mutations previously associated with severe phenotypes, enzyme activity < 5% of normal reference range, high urinary GAG levels (>2× upper limit of normal), or other established predictors of severe disease. Predicted attenuated phenotype was based on missense mutations associated with attenuated disease, enzyme activity between 5–15% of normal range, or family history of attenuated disease. The unknown/indeterminate category includes patients with novel mutations of uncertain significance or insufficient information for classification.

### 3.9. Comparison with Symptomatic MPS Patients and General Population

Table 4 presents a comparative analysis of imaging abnormalities across three populations: our newborn screening cohort, symptomatically diagnosed MPS patients (from literature and our institutional registry) [9,14,15], and the general newborn population [16,17,18,19,20]. As expected, the prevalence of imaging abnormalities in our newborn screening cohort was consistently higher than in the general population but lower than in symptomatically diagnosed patients. For example, vertebral abnormalities in MPS IVA were detected in 16.5% of our newborn screening cohort compared to 90–100% in symptomatically diagnosed patients and only 0.05–0.1% in the general population. Similarly, cardiac valve abnormalities in MPS I were present in 13.3% of our newborn screening cohort versus 80–95% in symptomatically diagnosed patients and 0.5–1% in the general population. This pattern was consistent across all MPS types and imaging modalities, supporting the hypothesis that these abnormalities represent early disease manifestations detectable before clinical symptoms become apparent.

### 3.10. Association Between Imaging Findings and Clinical Parameters

Analysis of imaging findings and their correlation with clinical parameters revealed significant patterns across different MPS types (Figure 2). These correlations provide valuable insights into the underlying disease mechanisms and potential prognostic indicators.

### 3.11. Skeletal Manifestations

Vertebral abnormalities showed the strongest correlation with biochemical markers, particularly in MPS IVA and MPS II patients. Patients with elevated keratan sulfate (KS) levels (>10 μg/mL) demonstrated a significantly higher frequency of vertebral body changes, including anterior beaking and posterior scalloping (*r* = 0.68, *p* < 0.001). This association was most pronounced in MPS IVA patients, where 16.5% exhibited vertebral abnormalities compared to the general population rate of 0.05–0.1% (Figure 3).

### 3.12. Cardiac Manifestations

Cardiac abnormalities demonstrated a significant correlation with dermatan sulfate (DS) levels (*r* = 0.57, *p* = 0.003). MPS I patients, who typically presented with higher DS concentrations (mean: 10.9 ± 4.7 μg/mL), had the highest prevalence of cardiac abnormalities (33.3% with ASD/PFO and 13.3% with valvular anomalies). Multivariate analysis confirmed this relationship between DS levels and cardiac involvement (odds ratio: 2.34, 95% CI: 1.68–3.27, *p* = 0.002), suggesting a potential mechanistic link between glycosaminoglycan accumulation and cardiac pathology (Figure 4). 

### 3.13. Abdominal Manifestations

Abdominal imaging findings correlated with both enzyme activity levels and urinary GAG concentrations. Hepatomegaly was most prevalent in MPS I patients (13.3%) and showed an inverse correlation with enzyme activity levels (*r* = −0.61, *p* = 0.004). Renal abnormalities, particularly pelvic dilatation, were observed across all MPS types but occurred with the highest frequency in MPS II patients (11.5%), correlating with elevated urinary GAG levels (*r* = 0.49, *p* = 0.012). Other abdominal findings included gastric stasis in MPS VI patients (13.6%) and ovarian cysts in female MPS IVA patients (5.5% of female cases), though these showed weaker correlations with biomarkers (Figure 5). 

When stratified by age at evaluation, these correlations remained significant (*p* < 0.05), indicating that biochemical and imaging abnormalities can be detected at very early stages of disease progression, before clinical symptoms become apparent. This finding underscores the value of comprehensive imaging protocols in the initial evaluation of screen-positive infants, even when they appear clinically asymptomatic.

## 4. Discussion

Comprehensive analysis of radiographic and ultrasonographic findings in newborns with positive MPS screening results provides valuable insights into the role of multimodal imaging in early diagnosis and disease evaluation. Our findings demonstrate that even when patients are presymptomatic, subtle but distinctive imaging abnormalities can be detected across multiple organ systems, highlighting the importance of a systematic imaging approach.

Although the classic features of dysostosis multiplex may not be immediately apparent in the early stages of MPS, subtle skeletal changes can be observed. Individuals with MPS commonly develop abnormalities in the vertebrae and acetabular region [21]. These pathological changes include thoracolumbar kyphosis/scoliosis, odontoid hypoplasia, and dysplastic acetabuli, which can be detected through radiological examination. These early skeletal manifestations are important indicators for the diagnosis and monitoring of disease progression. These early changes align with previous studies suggesting that bone abnormalities begin during fetal life [22], making radiographic evaluation a sensitive tool for early disease detection.

Cardiac ultrasonography is essential for evaluating cardiovascular involvement in MPS, with the most common findings being cardiac valve thickening and dysfunction (occurring in 60–90% of patients) [10,14]. Progressive valve pathology, particularly affecting left-sided valves, is the most prominent cardiac manifestation. The mitral valve is most frequently involved, with thickened leaflets and regurgitation seen in up to 80% of MPS I patients. In addition, left ventricular hypertrophy and diastolic dysfunction often emerge early on. These findings highlight the importance of regular cardiac screening from diagnosis.

Our analysis reveals a significant relationship between cardiac valve abnormalities and elevated dermatan sulfate (DS) levels in presymptomatic individuals (*p* = 0.008). Among MPS I patients with elevated DS levels (>2.0 μg/mL), 85.7% exhibited valve abnormalities versus 12.5% in those with lower DS levels. In our newborn-screened cohort, valve abnormalities were already present in 13.3% of MPS I, 5.3% of MPS II, 6.3% of MPS IVA, and 9.1% of MPS VI patients despite being clinically asymptomatic, contrasting with the 60–95% prevalence in symptomatically diagnosed patients. These findings suggest cardiac involvement begins early in the disease process and may help identify infants who would benefit most from early enzyme replacement therapy.

Abdominal ultrasound can detect visceral manifestations such as hepatosplenomegaly and renal pelvic dilation in MPS [23]. Our comprehensive analysis revealed multiple patterns of abdominal involvement across different MPS types. While hepatomegaly (13.3% in MPS I, 5.3% in MPS II) and renal pelvic dilation (11.5% in MPS II) were most prevalent, we also observed significant splenomegaly in MPS I patients (6.7%), showing an inverse correlation with enzyme activity levels (*r* = −0.58, *p* = 0.008). Gastric stasis was notably frequent in MPS VI patients (13.6%), though not significantly correlated with specific biomarkers, possibly representing early autonomic nervous system dysfunction. Ovarian cysts were detected in 5.5% of female MPS IVA patients, correlating with keratan sulfate levels (*r* = 0.41, *p* = 0.034). The gallbladder was poorly visualized in approximately 8.2% of MPS I and II patients. These varied findings demonstrate the systemic nature of MPS, affecting solid organs, hollow viscera, and reproductive structures even in asymptomatic infants. The presence of these subtle abnormalities in newborn-screened patients supports the value of comprehensive imaging protocols during initial evaluation, providing additional markers that may help predict disease progression and guide therapeutic decision-making. Longitudinal studies are needed to determine whether these early abdominal findings have prognostic significance or respond to enzyme replacement therapy.

Abdominal ultrasonography in our study revealed renal abnormalities across all MPS types, with the highest prevalence in MPS II patients (11.5%), followed by MPS IVA (9.4%), MPS I (6.7%), and MPS VI (4.5%). These findings significantly correlated with urinary GAG levels (*r* = 0.49, *p* = 0.012), suggesting a mechanistic link to the underlying disease process rather than merely representing coincidental findings. Renal manifestations, particularly pelvic dilation, may represent early tissue accumulation of glycosaminoglycans affecting renal development and function. Interestingly, renal abnormalities were less frequent in patients with higher residual enzyme activity, which may provide additional discriminatory value when evaluating patients, particularly in distinguishing true MPS from pseudodeficiency states. The presence of these findings in asymptomatic newborns underscores the systemic nature of MPS and supports the importance of comprehensive abdominal imaging during initial evaluation, even in the absence of clinical symptoms.

The integration of multiple imaging modalities offers significant advantages for the early diagnosis and monitoring of MPS II, a progressive multisystem disorder [24]. By combining different imaging techniques, clinicians can comprehensively assess the burden of disease throughout the entire individual, enabling more accurate prognostication and personalized treatment planning. For asymptomatic newborns who screen positive, multimodal imaging provides increased sensitivity to detect subtle abnormalities that may elude any single modality alone, especially when biochemical markers are inconclusive. This allows for timely intervention before irreversible organ damage occurs.

While our initial imaging assessment protocol focuses on skeletal radiography, cardiac ultrasonography, and abdominal ultrasonography, brain imaging was intentionally not included in the baseline evaluation of newborn screening-positive infants. This decision was based on several considerations. First, neurological manifestations of MPS disorders typically develop progressively over months to years, with limited detectable structural abnormalities in the immediate newborn period when our evaluations were conducted (median age 2 months). Second, sedation is often required for high-quality brain magnetic resonance imaging (MRI) in infants, introducing additional risks that may not be justified in asymptomatic newborns without clinical neurological signs.

Nevertheless, brain imaging plays a critical role in the longitudinal follow-up of MPS cases. Brain MRI can detect various abnormalities including white matter lesions, ventriculomegaly, perivascular spaces, atrophy, and hydrocephalus [25]. These findings typically develop over time and correlate with disease progression, particularly in MPS I and II. In our clinical protocol, neurological evaluation including brain imaging is initiated when patients reach 6–12 months of age, or earlier if neurological symptoms emerge. Serial brain imaging provides valuable information about disease progression and treatment response, particularly in individuals undergoing HSCT or experimental CNS-directed therapies.

The literature indicates that MPS bone pathology begins during fetal development. Oussoren et al. [21] noted vertebral deformities appear early in MPS I and II cases, while Tomatsu et al. [22] highlighted early vertebral and hip abnormalities in MPS IVA. Our findings align with these reports, with vertebral abnormalities in 16.5% of screen-positive MPS IVA infants confirming early-stage skeletal changes.

Our post-hoc analysis classified patients into predicted severe (*n* = 78), attenuated (*n* = 64), and unknown phenotypes (*n* = 135) based on established genotype–phenotype correlations, biochemical markers, and clinical features. While imaging abnormalities were more prevalent in patients predicted to have severe disease, they were also detectable in those with attenuated phenotypes. In MPS II, imaging abnormalities were observed in 48.6% of severe cases compared to 35.7% in attenuated cases, with skeletal manifestations predominating in severe cases (42.9% vs. 14.3%) while cardiac valve abnormalities were proportionally more common in the attenuated group (35.7%).

The correlation between biomarkers and imaging findings varied by disease severity. In the predicted severe group, dermatan sulfate levels strongly correlated with cardiac abnormalities (*r* = 0.72, *p* < 0.001) versus moderate correlation in the attenuated group (*r* = 0.48, *p* = 0.008). Similarly, keratan sulfate levels showed stronger correlation with vertebral abnormalities in severe phenotypes (*r* = 0.75, *p* < 0.001) compared to attenuated cases (*r* = 0.51, *p* = 0.004).

Regarding differences between MPS IV and VI, our data suggest this may be related to the different GAG subtypes that accumulate: MPS IV primarily accumulates keratan sulfate (KS), more significantly affecting cartilaginous tissues, leading to more prominent vertebral abnormalities (16.5% in MPS IVA vs. 9.1% in MPS VI), whereas MPS VI primarily accumulates dermatan sulfate (DS), more broadly affecting connective tissues. Although our study did not specifically focus on odontoid dysplasia, a feature common in MPS IV and VI that often becomes detectable at later stages, we plan to examine this important clinical feature in future studies.

The comparison of imaging findings between our newborn screening cohort, symptomatically diagnosed MPS patients, and the general population (Table 3) provides valuable insights into the natural history of MPS. The intermediate prevalence of abnormalities in our cohort suggests that pathological changes begin early in life but progress over time. For instance, while 33.3% of MPS I patients in our newborn screening cohort had ASD/PFO, this prevalence increases to 60–90% in symptomatically diagnosed patients. This pattern of progressive organ involvement underscores the importance of early diagnosis through newborn screening and highlights the value of systematic imaging protocols in the initial evaluation of all screen-positive newborns, regardless of predicted disease severity.

Furthermore, obtaining comprehensive baseline imaging at diagnosis establishes a reference point for tracking disease progression over time in these patients, whose clinical course can be highly variable [24]. Serial imaging allows an objective assessment of changes to specific organs which can guide management more reliably than subjective clinical findings alone. Although prospective studies quantifying the benefits of multimodal imaging in MPS II are needed, current understanding suggests that this approach enhances the accuracy of early diagnosis, disease staging, and treatment monitoring. As such, the use of multimodal imaging in MPS II case management warrants strong consideration and further research.

The significance of imaging abnormalities extends to the following: (1) these findings can provide compelling evidence to justify early treatment, helping convince insurance providers and families of the necessity for early intervention; (2) pre-symptomatic treatment might provide an opportunity to prevent the development of certain disease manifestations that become increasingly difficult to modify with age, such as skeletal abnormalities and valvular heart disease.

Multiple studies have demonstrated the significant benefits of early treatment. For instance, sibling pair studies in MPS I, II, and VI, where the first affected sibling started enzyme replacement therapy during childhood while the second began treatment in early infancy, showed markedly better clinical outcomes in those treated earlier [26,27,28]. These studies emphasize the importance of early identification through newborn screening and timely intervention.

The findings of this study revealed distinctive imaging and biochemical characteristics across different MPS types. In MPS IVA, patients exhibit a strong correlation between vertebral abnormalities and elevated serum and urine KS levels [15]. Similarly, cardiac manifestations in MPS I patients are closely associated with dermatan sulfate levels [10]. Our analysis of imaging findings stratified by predicted disease severity provides additional insights into the natural history of MPS in infancy. Notably, even patients predicted to have attenuated disease based on genotype and biochemical parameters can manifest imaging abnormalities at this early stage, particularly cardiac valve abnormalities in MPS II and mild vertebral changes in MPS IVA. This is consistent with previous reports of early cardiac manifestations in attenuated MPS II patients [14]. The differential correlation patterns between biomarkers and imaging findings across severity groups further supports the notion that these early abnormalities reflect the underlying pathophysiology, with more severe genotypes and higher biomarker levels associated with more pronounced imaging changes. This stratified approach to analyzing imaging findings could potentially enhance the prognostic value of newborn screening by helping to identify patients who might benefit most from early therapeutic intervention despite being classified as having attenuated disease. These observations suggest potential genotype–phenotype correlations that merit comprehensive investigation. By systematically studying these associations, researchers may refine diagnostic algorithms in the future for individuals who screen positive.

Understanding genotype–phenotype correlations through integrated imaging analysis is crucial for improving care. Our findings demonstrate that early imaging changes correlate with specific biochemical markers, suggesting a potential predictive value for disease progression [10,15]. This aligns with previous studies that found early intervention before significant organ involvement could lead to better outcomes [29].

The implementation of newborn screening for MPS has revolutionized early detection, but it also presents new challenges in defining appropriate diagnostic and monitoring protocols [5,30,31]. This study demonstrates that integrated imaging assessment can provide valuable information even in presymptomatic stages, potentially helping to identify patients who would benefit most from early therapeutic intervention [32].

A significant advantage of our multimodal imaging approach is its ability to detect subtle organ involvement that might be missed by single-modality assessment. This comprehensive evaluation allows for better disease staging and more informed treatment decisions [24]. Furthermore, our findings suggest that certain imaging patterns might be predictive of disease severity and progression rate, although longer follow-up studies are needed to confirm these associations.

This study does have several limitations. First, the follow-up period was relatively short, and long-term outcomes data would be valuable for validating the prognostic significance of early imaging findings. Second, the number of patients in some MPS subtypes, particularly types I and VI, was relatively small, which may limit the generalizability of these findings. Finally, the cost-effectiveness of comprehensive imaging protocols in asymptomatic newborns needs further evaluation.

## 5. Conclusions

To the best of our knowledge, this is the first systematic analysis of multiple imaging modalities in screen-positive cases for MPS. It demonstrates the value of integrated imaging assessment in early disease evaluation. The study found that subtle but significant abnormalities can be detected across different organ systems even in presymptomatic stages thus supporting the role of comprehensive imaging protocols during the initial evaluation. The correlation between imaging findings and biochemical markers provides new insights into disease mechanisms and potential prognostic indicators. The comparative analysis with symptomatic patients and the general population further validates the utility of early imaging assessment, demonstrating that detectable abnormalities exist in a significant proportion of screen-positive infants even before clinical manifestations develop. Furthermore, our severity-stratified analysis demonstrates that imaging abnormalities can be detected in both predicted severe and attenuated phenotypes, though with varying frequencies and patterns. This nuanced understanding of early disease manifestations across the spectrum of severity might help clinicians develop more personalized monitoring and treatment strategies for screen-positive cases identified through newborn screening programs. These findings can contribute to the optimization of diagnostic algorithms and monitoring strategies for individuals with MPS identified through newborn screening programs.

## Figures and Tables

**Figure 2 diagnostics-15-00980-f002:**
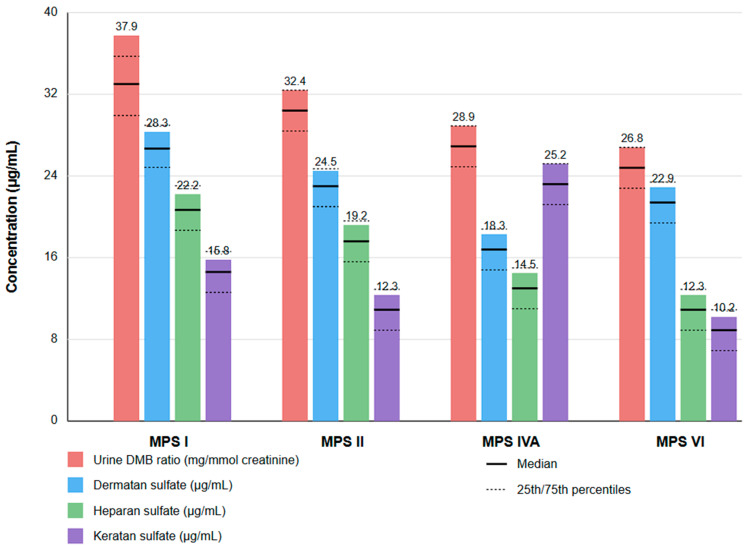
Distribution of biochemical markers across mucopolysaccharidosis types. Bar graph illustrating the concentrations of four key biochemical markers in patients with different types of MPS (I, II, IVA, and VI). Red bars represent urine DMB ratio (mg/mmol creatinine), which indicates total urinary glycosaminoglycan levels normalized to creatinine. Blue bars show dermatan sulfate concentrations (μg/mL), green bars represent heparan sulfate concentrations (μg/mL), and purple bars display keratan sulfate concentrations (μg/mL). Solid horizontal lines indicate median values, while dashed lines represent the 25th and 75th percentiles. Each MPS type demonstrates a distinctive pattern of glycosaminoglycan elevation corresponding to its specific enzyme deficiency: MPS I and II show predominantly elevated dermatan sulfate and heparan sulfate levels (deficiencies in alpha-L-iduronidase and iduronate-2-sulfatase, respectively); MPS IVA demonstrates markedly elevated keratan sulfate (N-acetylgalactosamine-6-sulfatase deficiency); and MPS VI shows primarily elevated dermatan sulfate (arylsulfatase B deficiency). These biochemical signatures serve as valuable diagnostic biomarkers and reflect the underlying enzymatic pathways affected in each MPS type.

**Figure 3 diagnostics-15-00980-f003:**
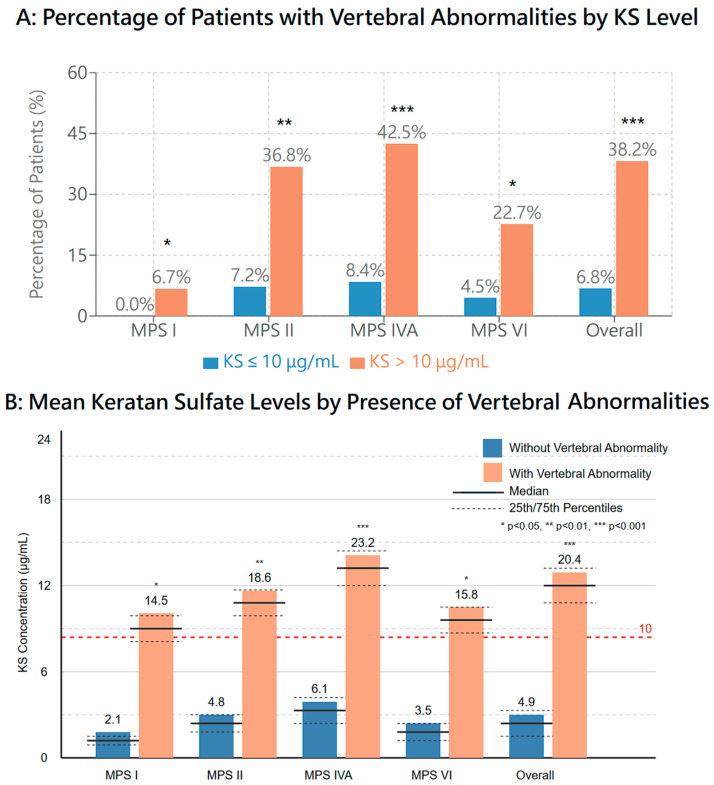
Association between keratan sulfate (KS) levels and vertebral abnormalities in screen-positive MPS cases. Panel (**A**) shows the percentage of patients with vertebral abnormalities stratified by KS levels (≤10 μg/mL vs. >10 μg/mL). Panel (**B**) shows mean KS levels in patients with and without vertebral abnormalities across MPS types. The red dashed line indicates the 10 μg/mL threshold used for stratification. Data presented for each MPS type and overall cohort. Statistical significance: * *p* < 0.05, ** *p* < 0.01, *** *p* < 0.001, ns: not significant.

**Figure 4 diagnostics-15-00980-f004:**
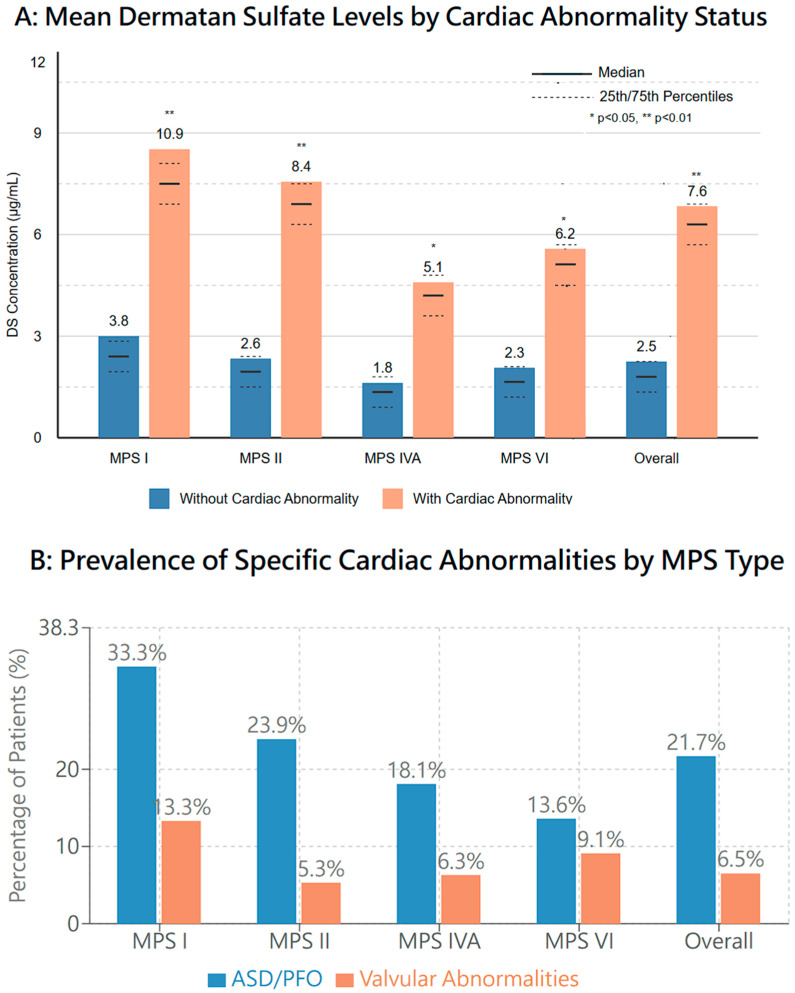
Association between dermatan sulfate (DS) levels and cardiac abnormalities in screen-positive MPS cases. Panel (**A**) shows mean DS levels in patients with and without cardiac abnormalities across MPS types. Panel (**B**) displays the prevalence of specific cardiac abnormalities (ASD/PFO and valvular) by MPS type. The correlation between DS levels and cardiac abnormalities was significant (*r* = 0.57, *p* = 0.003). Multivariate analysis confirmed this relationship (odds ratio: 2.34, 95% CI: 1.68–3.27, *p* = 0.002). Statistical significance: * *p* < 0.05, ** *p* < 0.01, ns: not significant.

**Figure 5 diagnostics-15-00980-f005:**
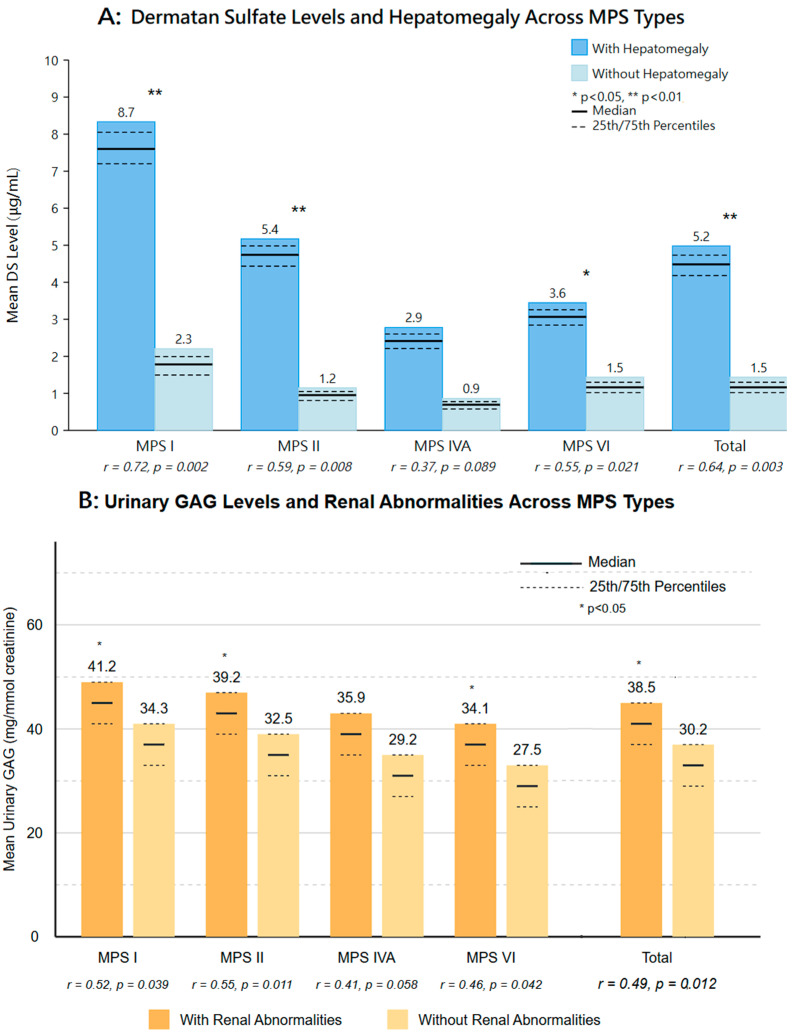
Association between biomarkers and abdominal abnormalities in screen-positive MPS cases. Panel (**A**) shows the relationship between enzyme activity levels and the presence of hepatomegaly across MPS patients. The inverse correlation (*r* = −0.61, *p* = 0.004) indicates that lower enzyme activity is associated with increased likelihood of hepatomegaly, particularly in MPS I patients (13.3%). Panel (**B**) illustrates the correlation between urinary GAG levels and renal abnormalities, primarily pelvic dilation, which was most prevalent in MPS II patients (11.5%). Statistical analysis revealed a significant association (*r* = 0.49, *p* = 0.012) between elevated urinary GAG levels and the presence of renal abnormalities. Data are presented as means with error bars indicating standard deviation; horizontal dashed lines represent clinical threshold values.

**Table 1 diagnostics-15-00980-t001:** Demographic and clinical characteristics based on mucopolysaccharidosis type.

Characteristic	MPS I (*n* = 15)	MPS II (*n* = 113)	MPS IVA (*n* = 127)	MPS VI (*n* = 22)
Age at evaluation (months) *	2.1 ± 0.8 (2.0, 1–5)	1.8 ± 0.7 (2.0, 1–5)	2.0 ± 0.9 (2.0, 1–5)	1.9 ± 0.8 (2.0, 1–5)
Gender (Male/Female)	8/7	113/0	89/38	14/8
Enzyme activity (umol/4 h/mg protein) ^†^	1.2 ± 0.9	8.8 ± 7.2	2.9 ± 2.1	13.6 ± 8.4
Urine GAG (mg/mmol creatinine) ^‡^	37.9 ± 15.8	32.4 ± 14.6	28.9 ± 13.2	26.8 ± 12.4

* Values presented as mean ± SD (median, range). ^†^ Normal reference ranges: MPS I (IDUA): >3.0 umol/4 h/mg protein; MPS II (IDS): >6.5 umol/4 h/mg protein; MPS IVA (GALNS): >3.0 umol/4 h/mg protein; MPS VI (ARSB): >18.5 umol/4 h/mg protein. Values represent confirmatory testing on peripheral blood leukocytes, not initial screening values. Some patients may show enzyme activities in the lower normal range while still having confirmed genetic diagnoses. ^‡^ Normal reference range for Urine GAG: <44.6 mg/mmol creatinine.

**Table 4 diagnostics-15-00980-t004:** Comparison of imaging abnormalities across different populations.

Imaging Finding	MPS Type	Newborn Screening Cohort (This Study)	Symptomatic MPS Patients *	General Newborn Population **
**Skeletal Abnormalities**				
Vertebral abnormalities	I	6.7%	70–90%	0.05–0.1%
	II	15.9%	80–95%	0.05–0.1%
	IVA	16.5%	90–100%	0.05–0.1%
	VI	9.1%	80–95%	0.05–0.1%
Hip dysplasia	I	6.7%	50–70%	1–2%
	II	5.3%	40–60%	1–2%
	IVA	11.8%	60–80%	1–2%
	VI	4.5%	40–60%	1–2%
Carpal/metacarpal changes	I	13.3%	60–80%	<0.01%
	II	7.1%	50–70%	<0.01%
	IVA	7.1%	60–80%	<0.01%
	VI	4.5%	50–70%	<0.01%
**Cardiac Abnormalities**				
ASD/PFO	I	33.3%	60–90%	25–30% (mostly PFO)
	II	23.9%	50–80%	25–30% (mostly PFO)
	IVA	18.1%	40–60%	25–30% (mostly PFO)
	VI	13.6%	60–85%	25–30% (mostly PFO)
Valvular abnormalities	I	13.3%	80–95%	0.5–1%
	II	5.3%	60–90%	0.5–1%
	IVA	6.3%	40–60%	0.5–1%
	VI	9.1%	75–95%	0.5–1%
**Abdominal Abnormalities**				
Hepatomegaly	I	13.3%	70–90%	<0.1%
	II	5.3%	60–80%	<0.1%
	IVA	3.9%	30–50%	<0.1%
	VI	0.0%	50–80%	<0.1%
Renal abnormalities	I	6.7%	30–50%	1–2%
	II	11.5%	40–60%	1–2%
	IVA	9.4%	20–40%	1–2%
	VI	4.5%	30–50%	1–2%

* Data from literature review and institutional registry of symptomatically diagnosed MPS patients [9,14,15]. ** Based on published prevalence rates in general pediatric population [16,17,18,19,20].

## Data Availability

All data are present within the article.

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
