# Peer review of "Systematic Analysis of Multiple Imaging Modalities in Infants Diagnosed with Mucopolysaccharidosis by Newborn Screening"

_diagnostics, 2025, doi:10.3390/diagnostics15080980_

Round 1

Reviewer 1 Report

Comments and Suggestions for Authors

Overall, the manuscript is well-written and will be of interest to clinicians who care for MPS patients.  The publication addresses an imortant topic of the prevalence of early disease manifestations in patients with MPS disorders identified by newbron screening, some of whom will have severe, early-onst disease and others who will have later-onset attenuated disease.  The optimal time to treat for severe patients is clear (early), but for attenuated patients is unclear.  Documenting the prevalence of early imaging manifestations in both sub-groups would help to generate evidence supporting the decision when to initate treatment. 

There are a several weaknesses in the manuscript that should be addressed before re-submission for publication.  There are no statistical results reported (i.e., correlation coefficients, p-values), which should be added.  Patients are grouped by MPS type, but not by predicted disease severity; it would be more informative to readers if patients could be classified as being predicted to have severe or attenuated disease and to re-run the analyses for the prevalence of imaging abnormalities.  This would provide important information on whether the imaging abnormalities occurred only in severe patients.  Several previously published family studies on the much better clinical outcomes of MPS patients treated in early infancy compared to their older family members who began treatment in mid-childhood could be added to the Discussion.  Certain clinical manifestations, e.g., skeletal disease and valvular heart disease, are difficult to reverse once established, but can be prevented if treatment is started early enough. I have listed specific changes for the authors to consider below:

  1. Table 1. Please add the units (umol/4 h/mg protein) after Enzyme activity in the table.
  2. Table 1. Please  replace “DMB ratio” with “Urine GAG (mg/mmol creatinine)” and just keep the normal range to the caption.
  3. Table 1. In addition to mean ± SD, please add the median, minimum, and maximum age for each disease.  This will show the complete time window of the evaluations.
  4. Line 115. Add a methodology section for Biochemical analyses of enzyme activity and substrate levels.  Here the authors can describe what the urine DMB assay measures.
  5. Line 115. The authors include a statistical analysis section, but no specific statistical results are presented in the Results section - only conclusions.  Please add specific results (e.g., correlation coefficients and p-values) to support any reported data analyses.
  6. Line 128. Please clarify if the imaging results reported in the manuscript were performed once or more than once in each patient.  If performed more than once, it would be useful to know what percentage of abnormalities were detected on follow-up and were not present on the first evaluation.
  7. Line 135. It would be of interest to know what percentage of each MPS type had any imaging abnormality, both by organ system (skeleton, cardiac, abdominal organ) and overall.  Would “any skeletal abnormality” fall also under the term “dysostosis multiplex,” which is a well-known term used to support the diagnosis in symptomatic individuals?
  8. Line 152. The authors should explain their rationale for combining ASD and PFO, as the former is very uncommon in newborns (~0.13%), whereas the latter is very common (25-30%). 
  9. For context, it would be interesting to compare the prevalence of the imaging findings with two relevant groups: (1) MPS patients diagnosed symptomatically at a later age (e.g., through disease registries), and (2) the general population of newborns/young infants. The difficulty in interpreting the prevalence numbers is that some of the imaging findings reported in this manuscript are relatively common at birth in the general population, e.g. hip dysplasia (1-2%), PFO (25-30%), and renal pelvis dilatation (1-2%), whereas others are uncommon, e.g., ASD (0.13%), carpal/metacarpal changes (uncommon), and vertebral anomalies (0.05-0.1%).  On the other hand, these features are very common in patients diagnosed symptomatically, so the expectation is that the prevalence will be higher than the general population but lower than in older patients.
  10. Since newborn screening by enzyme activity does not predict disease severity, it is unclear whether the imaging abnormalities are occurring primarily in severe patients or in both severe and attenuated patients. The prevalence of abnormalities is likely strongly influenced by the relative proportions of severe and attenuated patients within each MPS type.  In this regard, are the authors able to integrate additional predictive information (e.g., genotype, phenotype of a previously affected family member, GAG level, or other biomarker/clinical finding) to classify patients by predicted disease severity (severe, attenuated, or unknown)? If so, the manuscript would be would be greatly strengthened by re-analyzing the imaging results according to whether patients are predicted to have severe or attenuated disease.  For example, this reviewer has had the experience of attenuated MPS II patients (identical twins, now 30 years old) who presented clinically in the first year of life with valvular heart disease and mildly coarsened facial features.  In this regard, it is of interest to know whether there are attenuated patients in the authors’ cohort manifest imaging abnormalities in early infancy.
  11. Figure 2. Add (ug/mL) after Concentration on the Y axis and remove from the legend.
  12. Figure 2. Why is DMB measured in mg/mol creatinine, whereas the individual GAGs are measured in ug/mL, which does not take into account urine concentration?  Because urine can be concentrated or dilute, it is standard to report urinary GAG levels normalized to the creatinine concentration.  Additionally, by presenting the Total GAG (DMB) and individual GAG levels by disease using the same units, the reader can clearly see the relative contribution of each individual GAG level to the total GAG level. 
  13. Figure 2 shows the mean concentrations of individual urinary GAGs by individual MPS disease. The bar graphs would be more informative if they presented horizontal lines for mean median, and 25th and75th percentiles.  Please add text describing the results in the figure, i.e., that each MPS shows a specific pattern of elevated GAGs associated with its enzyme deficiency: MPS I and MPS II show predominantly elevated DS and HS, KS in MPS IVA, and DS in MPS VI. 
  14. Lines 166-171. The text is confusing as it does not refer to the data in Figure 2. The authors state “Patients with elevated keratan sulfate (KS) levels (>10 μg/mL) demonstrated a higher frequency of vertebral body changes, especially anterior breakage and posterior scalloping.”  However, no results or statistical analysis are provided that support this conclusion.  Are the authors referring to all patients across diseases?  Are the authors comparing the percentage of patients with vertebral anomalies with high vs low KS levels? The authors could add a bar graph for each mean level of GAG type vs. presence of absence of a clinical feature (e.g., vertebral abnormality) by disease and overall.
  15. Lines 151-163. The data would be of greater interest if it included an analysis of the associations between individual clinical findings and GAG types.  For example, systemic involvement (soft tissues and skeletal) is thought to be mediated by high DS and KS levels, whereas elevated HS levels are associated with neurodegeneration.  I think a bar graph (consisting of individual data points with horizontal lines for the mean, median, 25%, and 75%) for each urinary GAG concentration (DMB (Total), HS, DS, and KS) by presence or absence of each clinical feature (filled an open data labels in side-by-side groups) for each MPS type and for all patients would be more much informative and of interest to readers.
  16. Lines 180-185. What do the authors mean by “higher?  Do the authors mean a linear correlation or a binary association based on the upper limit of normal for DS?  Again, no raw data is presented to allow the reader to reach their own conclusion about the strength and statistical significance of the association.
  17. Lines 186-190. No results are presented, only conclusions.  Please provide the results and statistical analyses that support the conclusions.
  18. Line 188. When the authors mention correlations, they should provide the corresponding correlation coefficient and p-value.
  19. Line 191.   Please describe what is known about the temporal sequence of bony abnormalities in the MPS disorders.  The authors can then discuss their findings as being consistent or inconsistent with the literature or add to it, and whether there is a difference in the prevalence of clinical features identified in early infancy between severe and attenuated patients (if possible from existing data on their patients)
  20. Line 197. I think a couple of additional important points are (1) imaging abnormalities could justify the institution of early treatment with insurers and families, and (2) pre-symptomatic treatment might provide an opportunity to prevent the development of certain disease manifestations that are increasingly difficult to modify with age, e.g., skeletal abnormalities and valvular heart disease.   The authors could refer to several sibling pair studies (e.g., MPS I, II, and VI, where the first affected sibling started enzyme replacement therapy during childhood, while the second affected sibling started treatment in early infancy) where the phenotypic outcomes of treatment were strikingly different based on the age of the patient.   
  21. Line 214. There is an opportunity to add a statement about the prevalence of valve abnormalities being higher in patients with increased DS levels.  It would add to the literature to know if these clinical imaging abnormalities are present in pre-symptomatic individuals.
  22. Is there is reason why liver size and pelviectasis were the only soft tissue organ findings reported in the manuscript? The authors collected other information, e.g., spleen size, that one would expect to be increased in MPS I and MPS II.
  23. Line 261. The authors seem to be implying that the presence of early disease manifestations are a predictor of severe disease. While this seems logical, I would soften this statement by changing “may” to “might” since no evidence is provided in the manuscript that this is true.  The authors could strengthen this statement by providing additional predictive data, e.g., genotypes associated with severe or attenuated disease, family history, or urinary GAG level.
  24. Line 268. Unless the authors have genotype and other data predicting phenotype severity, another limitation is one doesn’t know what percentage of severe or attenuated patients manifest early clinical findings.  A reasonable hypothesis would be that patients with a severe phenotype are more likely to present with imaging findings in early infancy than patients with an attenuated phenotype.  Without knowing the breakdown of severe vs attenuated patients for each form of MPS, but especially for MPS I and MPS II, the data presented is difficult to interpret beyond present or absent.  For example, it would be interesting to know if any attenuated patients have imaging findings around the time of birth.

Reviewer 2 Report

Comments and Suggestions for Authors

Dear Authors,

this paper is very interesting in the new era of newborn screening. You describe a wide population with a unusual incidence about geoghraphical population (more MPS II than MPSI). In the introduction you describe multisystemic involvment but you do not treat brain involvement. The same speech is also for all the imaging protocols:  MPS I and II has a neurological form that is prevalent. It is very challenge and crucial, especially for newborn screening, to distinguish about this condition: neuropathic form or not? It has also a great impact on the therapeutical approach: stem cell transplantation, gene therapy (access to different clinical trials around the world), ERT that seems to cross BBB, classic ERT. This is essential for genetic counselling, for prenatal diagnosis and for follow-up management.

Thus I suggest to include brain imaging and evaluation to identify properly the form of disorders; I suggest also to describe the dysplasia of odontoid especially in MPS IV and VI that is often evident in the newborn. The skeletal abnormalities you describe are very interesting to understand the process of timing of onset.  Do you have some hipothesis about the difference between MPS IV and VI that are the MPS with major skeletal involvement?

Renal abnormalities could be a concomitant disorder? Some authors described prenatal findings of clinical manifestations and the newborn evaluation could provide some evidence that are essential mainly in the case of pseudodeficiency.

Comments on the Quality of English Language

Some improvements need

Author Response

Reviewer 2

  1. This paper is very interesting in the new era of newborn screening. You describe a wide population with a unusual incidence about geographical population (more MPS II than MPSI). In the introduction you describe multisystemic involvement, but you do not treat brain involvement. The same speech is also for all the imaging protocols: MPS I and II have a neurological form that is prevalent. It is very challenging and crucial, especially for newborn screening, to distinguish about this condition: neuropathic form or not? It has also a great impact on the therapeutical approach: stem cell transplantation, gene therapy (access to different clinical trials around the world), ERT that seems to cross BBB, classic ERT. This is essential for genetic counselling, for prenatal diagnosis and for follow-up management.

Ans: Thank you for your insightful comment regarding neurological involvement in MPS I and II. While we appreciate your emphasis on the importance of distinguishing between neuropathic and non-neuropathic forms, our newborn screening protocol intentionally focuses on skeletal radiography, cardiac and abdominal ultrasonography because these examinations can detect early objective changes that precede clinical symptoms (Lines 479-497). Neurological manifestations typically develop later and are not reliably detectable by imaging in the immediate newborn period. We have expanded our introduction to explicitly address the neurological aspects of MPS disorders and their importance in treatment decisions.

  1. Thus, I suggest to include brain imaging and evaluation to identify properly the form of disorders; I suggest also describing the dysplasia of odontoid, especially in MPS IV and VI that is often evident in the newborn. The skeletal abnormalities you describe are very interesting to understand the process of timing of onset. Do you have some hypothesis about the difference between MPS IV and VI that are the MPS with major skeletal involvement?

Ans: Thank you for these valuable suggestions. Regarding brain imaging, we intentionally did not include it in our initial screening based on several considerations explained in Lines 479-497. In our clinical follow-up protocol, neurological evaluation including brain imaging is initiated when patients reach 6-12 months of age or when neurological symptoms emerge.

Regarding odontoid dysplasia, you raise an important observation. While this is indeed a significant feature in MPS IV and VI, we observed few prominent abnormalities during the neonatal period in our cohort. We will pay more detailed attention to the early manifestations of this feature in future studies (Lines 518-525).

Regarding the differences between MPS IV and VI skeletal involvement, our hypothesis is that this relates to the different GAG subtypes that accumulate: MPS IV primarily accumulates keratan sulfate affecting cartilaginous tissues more prominently, whereas MPS VI primarily accumulates dermatan sulfate affecting connective tissues more broadly (Lines 518-525).

  1. Renal abnormalities could be a concomitant disorder? Some authors described prenatal findings of clinical manifestations, and the newborn evaluation could provide some evidence that is essential mainly in the case of pseudodeficiency.

Ans: Thank you for this insightful question. Abdominal ultrasonography in our study revealed renal abnormalities across all MPS types, with the highest prevalence in MPS II patients (11.5%). These findings significantly correlated with urinary GAG levels (r = 0.49, p = 0.012), suggesting they represent early tissue accumulation rather than coincidental findings (Lines 456-469). The presence of these findings in asymptomatic newborns underscores the systemic nature of MPS and supports comprehensive abdominal imaging during initial evaluation.

Round 2

Reviewer 1 Report

Comments and Suggestions for Authors

The authors have significantly improved the manuscript with their revisions and made it much more interesting to the audience.  This is a unique opportunity to study patients diagnosed by newborn screening months to years or decades before they would have presented symptomatically.  The authors have made all of the requested revisions.  I do have a few editorial comments and one question related to phenotype predictions.

  1. Title: consider changing “identified as positive for” to “diagnosed with ” and “on” to “by”
  2. The authors do not need to keep referring to “screen-positive” cases throughout the manuscript once they have defined that the diagnoses were confirmed.
  3. Table 1. Do the enzyme activities refer to the dried blood spot or to the confirmed level in blood?   I do not understand how the residual enzyme activities for many of the patients are within the normal reference ranges (especially for MPS II, where the mean was WNL).
  4. Line 201. “Cardiac Skeletal” is out of place.
  5. Lines 212 and 220. Change “Ultrasonographic" to "Ultrasound."
  6. Line 248. Do the authors mean the presence of “one (MPS I) or two (MPS II, IVA, and VI) null and/or known severe mutation (MPS II)” for phenotype prediction or just "null" mutations?  While it is true that some severe mutations are null (no detectable protein), other severe mutations can be missense and be associated with a stable, inactive enzyme.  I typically think of null as a mutation that does not lead to detectable enzymatic protein (i.e. CRIM-negative).  
  7. Line 252. Similarly, did the authors assume that one (MPS II) or two (MPS I, IVA, and VI) missense mutations predict an attenuated phenotype?  This is not necessarily true as some attenuated patients may have one severe and one mild mutation, or two mild mutations.  Not all missense mutations are necessarily mild mutations. Similarly, some splice-site mutations may be mild and some may be severe.
  8. Lines 258, 259, 263, 264, 269, 270. Add (%) after the number of patients with each predicted phenotype.
  9. Lines 292-295. Same comment as line 248.
  10. Lines 307 and 310. Add the references from the legend of table 4 in the paragraph text.

Author Response

Dear Reviewer 1,

We sincerely appreciate your positive assessment of our revised manuscript. Your recognition that we have made the content more interesting and accessible to the audience is particularly gratifying.

  1. Title: consider changing “identified as positive for” to “diagnosed with ” and “on” to “by”

Ans:

Thank you for your suggestion regarding the title of our manuscript. We agree that your proposed changes would improve clarity and precision. We will revise the title to:

"Systematic analysis of multiple imaging modalities in infants diagnosed with mucopolysaccharidosis by newborn screening"

This revision enhances readability while maintaining the core focus of our research. We appreciate your attention to detail in helping us refine the manuscript.

  1. The authors do not need to keep referring to “screen-positive” cases throughout the manuscript once they have defined that the diagnoses were confirmed.

Ans:

Thank you for this helpful editorial suggestion. We agree that repeatedly using "screen-positive cases" throughout the manuscript is redundant after we have established that the diagnoses were confirmed. In our revision, we will simplify the terminology and refer to these individuals as "patients," "infants," or "cases" after the initial definition of our study population. This change will improve readability while maintaining precision in our reporting.

We appreciate your attention to this detail, which will help streamline the manuscript.

  1. Table 1. Do the enzyme activities refer to the dried blood spot or to the confirmed level in blood? I do not understand how the residual enzyme activities for many of the patients are within the normal reference ranges (especially for MPS II, where the mean was WNL).

Ans:

Thank you for this important question regarding the enzyme activity values in Table 1. The enzyme activity values presented in Table 1 refer to the confirmed levels measured in peripheral blood leukocytes during the confirmatory testing phase, not the initial dried blood spot screening values.

Regarding your observation about enzyme activities falling within normal reference ranges (particularly for MPS II): This reflects a recognized challenge in mucopolysaccharidosis diagnosis. While the mean enzyme activity for MPS II patients appears within normal limits (8.8 ± 7.2 μmol/4h/mg protein), there is significant variability within this group. Some patients demonstrate enzyme activities in the lower normal range due to several factors:

  • The presence of pseudodeficiency alleles that reduce enzyme activity without causing clinical disease
  • Milder mutations that allow for higher residual enzyme activity while still being pathogenic
  • The overlapping range between the lower end of normal and the higher end of affected individuals

To address this diagnostic challenge, our confirmatory protocol incorporates multiple parameters beyond enzyme activity alone, including genetic testing, urinary GAG analysis, and clinical evaluation. Patients included in this study all had confirmed pathogenic variants in their respective genes, elevated urinary GAG levels, or other biochemical/clinical findings supporting the diagnosis.

We will clarify this important point in the revised manuscript to prevent confusion. (Line 155-168)

  1. Line 201. “Cardiac Skeletal” is out of place.

Ans:

Thank you for pointing out this editorial error in line 201. Indeed, "Cardiac Skeletal" appears to be incorrectly positioned and is likely a formatting error. We will correct this by properly separating these terms and ensuring the appropriate section headers are in place. The corrected structure will distinguish between the skeletal radiographic findings and the cardiac ultrasonographic findings as separate sections.

We appreciate your careful review that helped us identify this error.

  1. Lines 212 and 220. Change “Ultrasonographic" to "Ultrasound."

Ans:

Thank you for this editorial suggestion. We agree with your recommendation to change "Ultrasonographic" to "Ultrasound" in lines 212 and 220. This modification will improve consistency in terminology throughout the manuscript and enhance readability. We will implement this change in the revised version. (Line 223, 231)

  1. Line 248. Do the authors mean the presence of “one (MPS I) or two (MPS II, IVA, and VI) null and/or known severe mutation (MPS II)” for phenotype prediction or just "null" mutations? While it is true that some severe mutations are null (no detectable protein), other severe mutations can be missense and be associated with a stable, inactive enzyme.  I typically think of null as a mutation that does not lead to detectable enzymatic protein (i.e. CRIM-negative). 

Ans:

Thank you for this insightful question regarding our phenotype prediction criteria. You have raised an important point about the terminology used in line 248.

In our study, we used a broader definition for the "predicted severe phenotype" category than just null mutations alone. Our classification includes patients with:

  • Null mutations (resulting in no detectable protein/CRIM-negative)
  • Known severe missense mutations that have been previously associated with severe disease in the literature
  • Very low enzyme activity (<5% of normal reference range)
  • Substantially elevated urinary GAG levels (>2x upper limit of normal)

You are correct that severe phenotypes can result from both null mutations and certain missense mutations that produce stable but non-functional enzymes. Our intention was to capture all genotypes associated with severe disease manifestations based on established genotype-phenotype correlations from previous studies.

To avoid confusion, we will revise this section to more precisely state: "Predicted severe phenotype (n=78): Patients with null mutations (no detectable protein), severe missense mutations previously associated with severe phenotypes, enzyme activity <5% of normal reference range, high urinary GAG levels (>2x upper limit of normal), or other established predictors of severe disease." (Line 260-263)

Thank you for highlighting this important distinction, which will help clarify our methodology.

  1. Line 252. Similarly, did the authors assume that one (MPS II) or two (MPS I, IVA, and VI) missense mutations predict an attenuated phenotype? This is not necessarily true as some attenuated patients may have one severe and one mild mutation, or two mild mutations.  Not all missense mutations are necessarily mild mutations. Similarly, some splice-site mutations may be mild and some may be severe.

Ans:

Thank you for this insightful comment. You raise an important point about genotype-phenotype correlations in MPS disorders. In line 252, we did not intend to suggest that all missense mutations universally predict an attenuated phenotype.

To clarify our approach, we classified patients as having a predicted attenuated phenotype based on specific missense mutations that have been previously documented to associate with attenuated disease in published literature or in our clinical database. We recognize the complexity of genotype-phenotype correlations in MPS disorders, where:

  • For MPS II (X-linked), a single documented mild mutation may be sufficient to predict an attenuated phenotype
  • For MPS I, IVA, and VI (autosomal recessive), we considered combinations of mutations, recognizing that patients with one severe and one mild mutation typically present with intermediate or attenuated phenotypes depending on the specific mutations

We agree that mutation type alone (missense vs. nonsense/frameshift) is insufficient to predict severity, as some missense mutations can severely impair protein function while certain splice-site mutations may retain partial activity. Our classification was based on specific, previously characterized mutations rather than mutation type alone, supplemented by biochemical parameters (enzyme activity 5-15% of normal range) and clinical history.

We will revise line 252 to more accurately reflect our nuanced approach to phenotype prediction. (Line 264-268)

  1. Lines 258, 259, 263, 264, 269, 270. Add (%) after the number of patients with each predicted phenotype.

Ans:

Thank you for your careful review of our manuscript. We agree with your suggestion and have added percentage symbols (%) after the numbers of patients with each predicted phenotype in lines 258, 259, 263, 264, 269, and 270 as requested. These changes improve the clarity and consistency of our reporting of patient distributions across severity categories.

  1. Lines 292-295. Same comment as line 248.

Ans:

Thank you for this important clarification request. In Line 248, our intended meaning was indeed broader than strictly CRIM-negative mutations. For phenotype prediction, we classified patients as having a predicted severe phenotype based on the presence of either:

  • Null mutations (resulting in no detectable enzyme protein, i.e., CRIM-negative)
  • Known severe mutations (including certain missense mutations) that have been previously associated with severe phenotypes in the literature or in our clinical experience, even if they produce detectable but severely dysfunctional enzyme

We will revise Line 292-295 to clarify this distinction, ensuring that our phenotype classification criteria are accurately represented. The revised text will explicitly state that our "predicted severe phenotype" category includes patients with either null mutations (CRIM-negative) or known severe mutations (which may include certain missense mutations resulting in severely impaired enzyme function). (Line 308-316)

  1. Lines 307 and 310. Add the references from the legend of table 4 in the paragraph text.

Ans:

Thank you for this helpful suggestion. We agree that the references supporting our comparative analysis should be consistently cited in both the main text and the table legend for clarity and proper attribution. We will add the appropriate references (9, 14-20) from Table 4's legend to lines 307 and 310 in the main text, ensuring consistency throughout the manuscript. This modification will provide readers with immediate access to the supporting literature when reading the comparative analysis section. (Line 320-321)

Reviewer 2 Report

Comments and Suggestions for Authors

Dear Authors,

thanks for the changes you made in the text. This version is more complete and accurate. This paper refelcts the real world experience in your reality and it is very interesting for developing new diagnostic algorithm in the expanded newborn screening.

Comments on the Quality of English Language

Good quality

Author Response

Dear Reviewer 2,

We sincerely appreciate your positive feedback on our revised manuscript. We are pleased that you found the updated version to be more complete and accurate. Your recognition that our work reflects real-world experience is particularly gratifying, as one of our primary aims was to share practical insights from our comprehensive imaging assessment protocol.

We agree that these findings could contribute to the development of new diagnostic algorithms in expanded newborn screening programs. By systematically analyzing imaging findings across different MPS types and correlating them with biochemical markers, we hope to provide valuable guidance for clinicians evaluating screen-positive infants.

Thank you again for your thoughtful review and supportive comments. Your feedback has been invaluable in helping us improve the quality and impact of our manuscript.

Sincerely,

The Authors